# UMI on Legs: Making Manipulation Policies Mobile with Manipulation-Centric Whole-body Controllers

Huy Ha[*,1,2], Yihuai Gao[*,1], Zipeng Fu[1], Jie Tan[3], and Shuran Song[1,2]

[1]*Stanford University,* [2]*Columbia University,* [3]*Google DeepMind*

**Abstract:** We introduce UMI-on-Legs, a new framework that combines real-world and simulation data for quadruped manipulation systems. We scale task-centric data collection in the real world using a hand-held gripper (UMI), providing a cheap way to demonstrate task-relevant manipulation skills without a robot. Simultaneously, we scale robot-centric data in simulation by training whole-body controller for task-tracking without task simulation setups. The interface between these two policies is end-effector trajectories in the world frame, inferred by the manipulation policy and passed to the whole-body controller for tracking. We evaluate UMI-on-Legs on prehensile, non-prehensile, and dynamic manipulation tasks, and report over 70% success rate on all tasks. Further, we conduct in-the-wild, OOD evaluation for both policies in 5 unique locations (indoor/outdoor, deformable/hard ground) and report 60% success rate. Lastly, we demonstrate the zero-shot cross-embodiment deployment of a pre-trained manipulation policy checkpoint from prior work, originally intended for a fixed-base robot arm, on our quadruped system. Please checkout our website for robot videos, code, and data: https://umi-on-legs.github.io/

**Keywords:** Manipulation, Visuo-motor policy, Whole-body Control

**Task-Centric Human Demonstrations in the Real World**

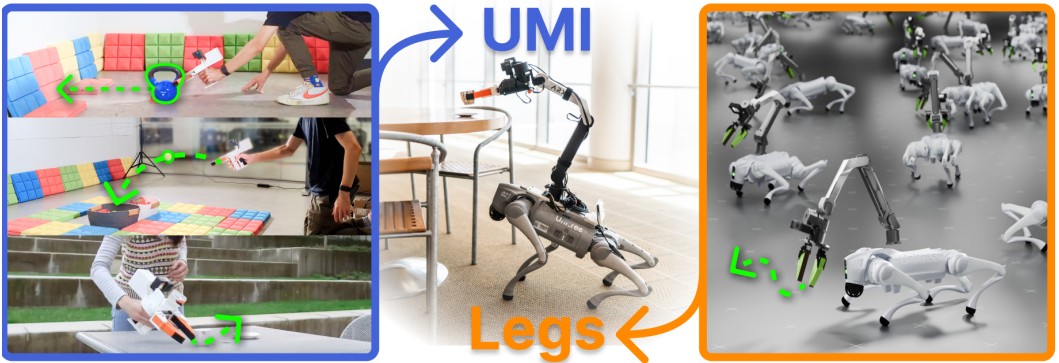

**Robot-Centric Reinforcement Learning in Simulation**

Fig 1: **UMI-on-Legs**. We achieve fully autonomous, expressive, real-world manipulation skills on quadrupeds by combining real-world demonstrations using hand-held grippers (left) and simulation-trained whole-body controllers (right). Our framework allows the porting of *existing* "table-top" manipulation policies to mobile manipulation while enhancing mobility and power from the quadruped's legs.

## 1 Introduction

Collecting robot data in simulation versus in the real world are two distinct pathways for scaling up robot learning approaches, each with its challenges. Real-world robot teleoperation allows for direct human demonstration of new tasks. However, it is often bottlenecked by the *robot's physical hardware* – in addition to the cost and safety concerns, the involvement of robot hardware usually makes the collected data robot-specific (e.g., uses body commands) or even irrelevant whenever the robot's physical form changes. In contrast, simulation provides safe exploration and infinite resets for any new robot design. However, it is often bottlenecked by the *task diversity* – accurately simulating diverse objects and their dynamics and defining all tasks and their associated rewards remains a significant challenge.

---

∗ Indicates equal contribution

8th Conference on Robot Learning (CoRL 2024), Munich, Germany.

In this work, we introduce **UMI-on-Legs**, a quadruped manipulation system that demonstrates an alternative way to combine real-world and simulation data:

- **Scaling task-centric data in the real world without a robot:** We propose using a hand-held gripper (i.e., UMI [1]) to demonstrate task-centric data without requiring robot hardware. We model this manipulation data using a diffusion policy [2] that is agnostic to the robot embodiment beyond the end-effector. By allowing human operators to collect the data directly, we can easily capture a diverse set of challenging tasks, bypassing the need for real-world robot trial-and-error.

- **Scaling robot-centric data in simulation without task simulations:** We train a whole-body controller (WBC) in massively parallelized simulation [3–7], whose sim2real transfer to a wide range of terrain conditions [7–9] is a well-studied problem. By training control policies to track end-effector trajectories instead of interacting with objects, we can bypass the difficulty of setting up realistic manipulation assets, engineering task rewards, and modeling object dynamics.

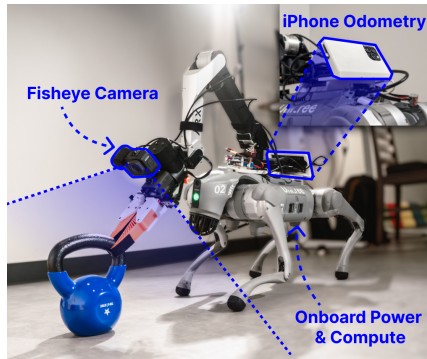

Fig 2: **In-the-wild Manipulation, on Legs**. Featuring a self-contained iPhone odometry and onboard power/ compute, our mobile manipulation system is complete for in-the-wild manipulation.

A key design choice in the framework is the interface between the two policies: it must be simple enough for non-experts to use, yet expressive enough for complex manipulation tasks. Most prior legged manipulation systems use body velocity commands with single-step, body-frame end-effector targets [8, 10–12], which are embodiment-specific and insufficient for dynamic manipulation. We propose using *end-effector trajectories in the world-frame*, where the manipulation policy infers these from gripper-centric image observations. The whole-body controller then converts these trajectories into embodiment-specific controls, enabling cross-embodiment generalization of the manipulation policy and providing preview for anticipating motions for the controller.

To support system deployment, we present an accessible, real-time odometry solution, based on the Apple's ARKit. Instead of assuming a tightly integrated and precisely calibrated/synchronized system [13–15], our odometry approach takes the self-contained, compact form factor of an iPhone.

We tested UMI-on-Legs with complex tasks requiring prehensile actions (pick & place), non-prehensile whole-body actions (weight pushing), and dynamic actions (tossing) in challenging scenarios with on-board sensing and in-the-wild generalization. Thanks to our interface design, our asynchronous bi-level policy formulation robustly executes these tasks, achieving over 70% on all tasks. In summary, the main contributions of this work are:

- **UMI-on-Legs**, a framework for combining real-world and simulation data, which enables learning expressive manipulation skills on mobile robots using no real-world robot data. Namely, real-world data collected without any robots [1] can be distilled into manipulation skills for different mobile robot platforms using robot-specific controllers trained entirely in simulation.

- Within this framework, we propose a **manipulation-centric whole-body controller**, using an end-effector trajectory interface. This simple interface enables *zero-shot cross-embodiment deployment* of existing manipulation policies while being expressive enough to represent complex manipulation skills.

- A **real-world deployment system** featuring a real-time, robust, and accessible odometry approach for in-the-wild task-space tracking, addressing a common bottleneck in achieving fast and precise manipulation in mobile manipulation systems.

## 2 Related Work

**Mobile Pick-and-Place Systems.** Prior works have shown that pick-and-place quadrupeds trained in simulation can be deployed in real-world environments using object detectors for visual domain transfer [12, 16]. Wheeled systems demonstrate more complex manipulation tasks using carefully engineered primitives [17–21], enabling perception and planning [17, 20, 22] with foundation models [23–26], online learning [18, 19], or trajectory optimization [21]. Recently, Mobile-ALOHA [27] proposed a wheeled-bimanual platform for demonstration collection. However, these systems rely on quasi-static manipulation or heavy bases with a low center of mass.

**Learning-based Controllers for Robot-specific Teleoperation.** Reinforcement learning [28] (RL) with massively parallel simulators [3, 7] is the dominant approach for learning dynamic locomotion [6, 29–31] and manipulation skills [32–34] in legged robots. While domain adaptation methods [7–9] allow deployment in different environments, mobile-manipulation on quadrupeds still typically requires human teleoperation/AprilTags [35] during deployment for teleoperation [8, 10, 11]. Fully-autonomous deployment is limited due to the high cost of collecting real world robot data, which depends on robot-specific commands and requires the robot's physical presence. Recently, He et al. combines behavior cloning with an RL controller, but their approach relies on robot-specific demonstrations generated heuristically in simulation, limiting scalability, expressivity, and usability. Furthermore, their system is constrained by a tethered setup with external cameras for April Tag tracking.

**Cross-embodiment Manipulation.** In pursuit of robust and generalizable manipulation, behavior cloning works has innovated on policy architectures [2, 37, 38], sensor placements [37], action spaces [38], and data considerations like quality, size, and cost [37, 39–43]. Key design choices have emerged, including pre-trained visual encoders [1, 25, 39], diffusion-based action decoding [2, 39, 44], and end-effector sequence prediction [1, 38, 44]. Recent work [1, 45] shows zero-shot cross-embodiment visuo-motor policy transfer using only ego-centric visual inputs. However, by predicting target pose sequences, these approaches assume precise low-level controllers for tracking.

## 3 Method: Universal Manipulation Interface on Legged Robots

UMI-on-Legs (Fig 3) comprises of two main components: (1) a high-level diffusion-based manipulation policy [2] which takes as input wrist-mounted camera views and outputs sequences of end-effector pose targets into the future in the camera frame, and (2) a low-level whole-body controller which tracks end-effector pose targets by outputting joint position targets for both the legs and arm. We train the manipulation policy using data collected in the real world using UMI [1], a hand-held gripper data collection device and train the WBC entirely in simulation using massively parallelized simulators [3].

Our choice of using world-frame end-effector trajectories as the interface has the following benefits:

- **Intuitive demonstration**: Using end-effector trajectories instead of robot-specific low-level actions, we allow intuitive task demonstration by non-expert users using hand-held devices like UMI [1].

- **High-level intention from preview horizon**: With the preview horizon of future targets, the whole body controller can anticipate upcoming movements. For instance, the robots should brace accordingly if a high-velocity toss is coming up. Meanwhile, if the target is moving within the arm's reach range, the body should lean instead of taking a step, which risks shaking the end effector.

- **Precise and stable manipulation in the world-frame**: Unlike most of legged manipulation systems that uses body-frame tracking, our controller tracks action in the task-space (Fig 4) that is persistent regardless of base movement, enabling precise and stable manipulation.

- **Asynchronous multi-frequency execution:** The interface defines a natural inference hierarchy that allows a low-frequency manipulation policy (1-5Hz) to coordinate with a high-frequency low-level controller (50Hz) for handling drastically different sensor and inference latencies.

- **Compatible with any trajectory-based manipulation policy**: Our interface supports the plug-and-play of any trajectory-based manipulation policy [1, 38, 39, 43–45]. With the rise of policies trained on diverse datasets [39–41, 45, 46], our manipulation-centric WBC can accelerate the porting of existing "table-top" manipulation skills to "mobile" manipulation.

### 3.1 Manipulation Policy with Behavior Cloning

Following the default configuration from Chi et al., we use a U-Net [47] architecture diffusion policy [2] (Fig 3a) with the DDIM scheduler [48] and a pre-trained CLIP vision encoder [25] to map from a context of visual observations to end-effector pose and gripper command trajectories. We use a longer action horizon of 64 to provide more information into the future for the low-level controller. For our cup rearrangement task, we directly use UMI [1]'s cup rearrangement checkpoint. For the pushing and tossing task, we collect data and train diffusion policies from scratch. Tasked with processing high-dimensional visual data, this network runs much slower than the controller. However, its output pose trajectory can be densely interpolated to get inputs for the controller, which outputs position targets for 12 leg joints and 6 arm joints. Meanwhile, its output gripper command trajectory can be interpolated can fed directly into the gripper joint.

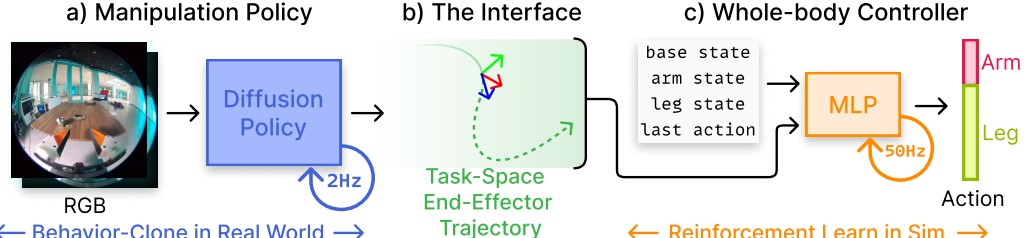

Fig 3: **Method Overview**. Our system takes as input RGB images from a GoPro and infers a camera-frame end-effector trajectory using a diffusion policy (a), trained using real-world UMI demonstrations. We transform this trajectory into the task-space, and use it as the interface to the WBC. This controller (c) outputs joint position targets at 50Hz, which PD controllers subsequently tracks.

## 3.2 Whole-body Controller with Reinforcement Learning

To track end-effector trajectories predicted from the manipulation policy, we propose to train a whole body controller with reinforcement learning in simulation to infer arm and leg joint targets. Notably, setting up a simulation to track these manipulation end-effector trajectories does not require setting up the manipulation task and environment. This design greatly alleviates a key bottleneck in using simulation data.

**World-frame Trajectory Tracking of Manipulation Trajectories.** Prior works [8, 10–12] train WBCs by sampling end-effector poses in the *body-frame*, simplifying policy optimization but failing to teach whole-body coordination to compensate for base movements (Fig 4b). This limitation is more pronounced when the arm's momentum affects the base, especially with lightweight systems or dynamic movements. In contrast, we train our controller to track trajectories in the world-frame (Fig 4a), teaching the robot to maintain its gripper pose by compensating for body movements. We use task trajectories from UMI [1], which are more representative of real manipulation tasks. In § 4, we show our controller generalizes across tasks without sacrificing tracking performance. Deployment details are in § 3.3.

**Observation space**. The observation space includes robots' 18 joint positions and velocities, the base orientation and angular velocity, the previous action, and the end-effector trajectory inferred by the manipulation policy. We represent end effector pose using a 3D vector for the position and the 6D rotation representation [49]. We densely sample target poses from -60ms to 60ms at 20ms interval relative to the current time, which informs the controller of current velocity and acceleration. In addition, we include the target 1000ms into the future, which helps the controller prepare to take foot steps if necessary.

**Rewards**. The task objective rewards the policy for minimizing the position error $\epsilon_{pos}$ and orientation error $\epsilon_{orn}$ to the target pose as follows: $\exp(-(\frac{\epsilon_{pos}}{\sigma_{pos}} + \frac{\epsilon_{orn}}{\sigma_{orn}}))$, where $\sigma$ represents scaling terms based on precision requirements. This formulation entangles position and orientation, resulting in more ideal behavior compared to treating them separately, which often leads to high precision in only one aspect. We also found that implementing a $\sigma$ curriculum for both position and orientation is essential for enabling exploration early on while ensuring high precision with further training. In addition, we incorporate common regularizations [7–9, 12], as detailed in the supplementary material.

**Policy network architecture.** We train a multi-layer perceptron (MLP) controller which maps from observations to target joint positions for both the legs (12 DOF) and the arm (6 DOF) (Fig 3c). A forward pass for this controller takes ≈0.06ms on the robot's on-board CPU, and is called at 50Hz during deployment. The joint position targets are tracked using separate PD controllers for the legs and the arm.

## 3.3 System Integration

**Robot system setup.** The robot system (Fig. 2) consists of a 12-DOF Unitree Go2 quadruped robot and a 6-DOF ARX5 robot arm, both powered by the battery of Go2. We customize the ARX5 arm with Finray grippers and a GoPro to match the UMI gripper [1]. Our whole-body controller runs on the Go2's Jetson while our diffusion policy inference runs on a separate desktop's RTX 4090 over an internet connection. We mount an iPhone for pose estimation and connect it to the Jetson through an Ethernet cable.

**Sim2Real Transfer.** To improve robustness, we apply random push forces and randomize joint friction, damping, contact frictions, body/arm masses, and centers of mass. Domain randomization of the end-effector's mass and center of mass simulates the effects of grasped objects without explicitly modeling them. Modeling a

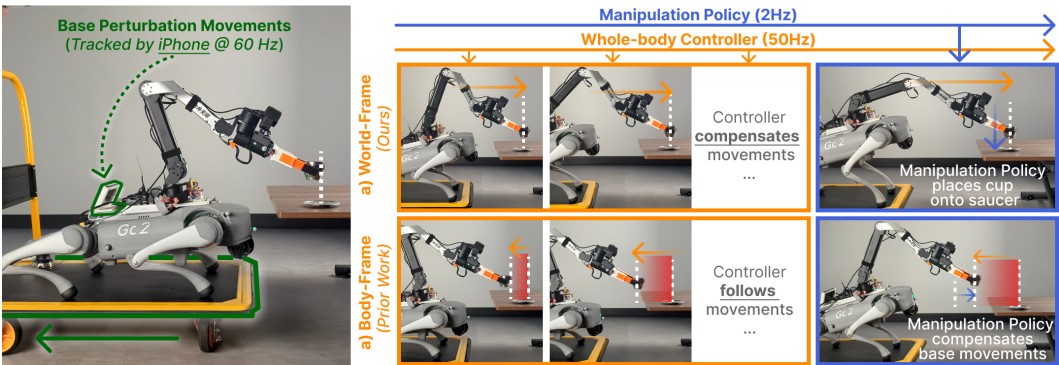

Fig 4: **World- v.s. body-frame tracking**. Our whole-body controller (WBC) learns to track the target trajectory in **world-frame (a)**, effectively *compensating* base perturbations and, therefore, frees up the manipulation policy to focus on making task progress. In contrast, most of existing WBCs use **body-frame tracking (b)**[8, 10–12], trained to *follow* base perturbations. In effect, they defer world-frame tracking responsibilities to the low-rate manipulation policy and fail to react quickly to body perturbations.

20ms control latency during training was essential. Additionally, to account for odometry noise, we randomly transport the robot every 20 seconds mid-episode. More details are provided in the appendix.

**Accessible real-time odometry.** The lack of real-time onboard task-space tracking is a common limitation in prior quadruped manipulation works. By assuming external tracking with motion capture [32] and/or AprilTags [8, 11, 36], their systems can't be deployed fully-autonomously in the wild. In our system, we address this drawback with an iPhone, mounted on the robot's base. Our choice of the rear mounting site avoids adding extra weight on the robot arm, prevents arm-phone collisions, and minimizes motion blur and visual occlusion. In contrast to many existing robust, real-time odometry solutions [13–15], our odometry solution has a self-contained, compact form-factor and uses only ubiquitous consumer electronic devices.

**Asynchronous Transform Book-keeping.** During deployment, the manipulation policy, WBC, and iPhone pose estimator run asynchronously. Let $t$ be the current timestep and $T$ the action horizon of the manipulation policy. The iPhone ARSession's frame serves as the world-frame, initialized when the iOS app opens. For each camera-frame action trajectory $a_{t+T}$ from the manipulation policy, we transform it into the world-frame using the camera-pose recorded at time $t$. This pose is derived from the iPhone's timestamped world-frame pose and joint state forward kinematics. For all timesteps $t' \in [t, t+T]$, the controller densely interpolates the end-effector trajectory then transform it into the gripper's frame for controller observations.

# 4   Experiments

We designed a series of experiments in simulation and the real world to validate our key design decisions, which aims to answer the following key questions:

- **Capability.** Is our system UMI-on-Legs able to learn complex and challenging manipulation skills such as whole-body dynamics actions (e.g., tossing)?
- **Robustness.** Can our whole-body controller handle unexpected external perturbations and object dynamics, encountered during manipulation?
- **Scalability.** Can we use cross-embodiment manipulation data collected entirely absent of the quadruped?

**Metrics.** In simulation, we report average position error (cm), orientation error (radians), survival (%), and electrical power usage (kW) over 500 episodes. Here, survival is 1 for the episode only if the policy lasts for 17 seconds without a terminal collision (any robot link contact that is not the robot's feet or gripper). In real-world experiments, we report average success rate over 20 episodes unless specified otherwise.

**Training Data.** We train a WBC and a manipulation policy for each task on only data from those tasks: 100 trajectories for tossing, 200 for pushing, and 1000 for cup rearrangement. An exception is the cup task manipulation policy, where we directly use the public checkpoint from UMI [1], and only train a WBC for this task. To investigate the effects of adding more task trajectories, we finetuned "Ours (Tossing)" on 300 trajectories, 100 from tossing, pushing, and cup rearrangement each, leading to "Ours (Unified)".

**Comparisons.** We ablate our design choices for the WBC in Table 1. Starting with our approach, Ours (Tossing), we remove preview (i.e., trajectory observations), world-frame tracking formulation, and

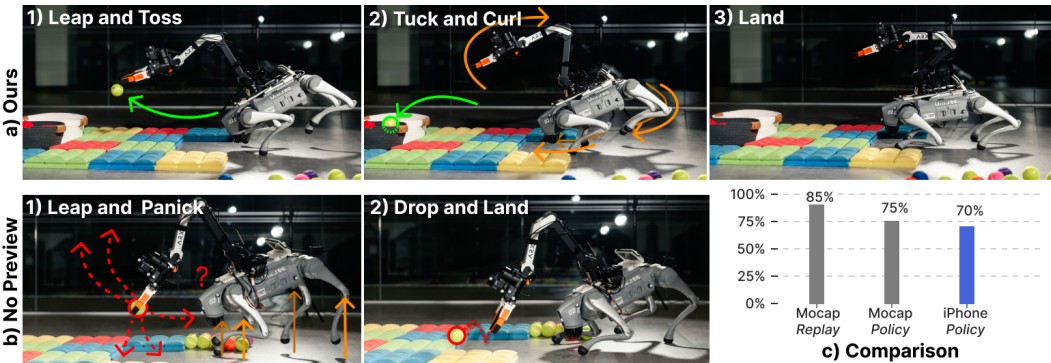

Fig 5: **Dynamic Tossing Requires Whole-body Coordination.** Our controller (top row) effectively executes a reliable tossing strategy, despite limited arm strength and body inertia, in three stages. First, as the arm accelerates forward, the back legs lift, creating a leap and toss motion (1a, green). To avoid falling forward, the robot tucks its arms and legs inwards (2a, orange), generating backward torque that shifts the front legs' contact points ahead of its center of mass for a soft landing (3a). In contrast, the controller without preview information (bottom row) attempts to leap after the fast target acceleration but lacks foresight, resulting in a dropped ball. Check out our robot videos at our website!

manipulation trajectory training trajectories. Removing all three simultaneously leads to a baseline similar to DeepWBC [8]. Finally, to investigate the efficacy of our iPhone-based odometry system, we compare against an OptiTrack motion capture system in all our real-world experiments, which serves as the oracle odometry. For real-world tossing, we also show the no-preview baseline's behavior in real (Fig. 5).

### 4.1 Capability: Whole-Body Dynamic Tossing

Our first experiment aims at validating the system's capability, i.e., whether our policy interface can capture complex manipulation skills. For this, we chose a particularly challenging task – **dynamic tossing**. The robot, with the center of its body 150cm away from and facing the target bin, must toss pre-grasped balls towards a target bin 90cm away, where a trial is successful if it lands within 40cm of bin's center.

**Task difficulties.** Most of existing tossing systems assume a carefully designed primitive [17, 50]. Recently, Chi et al. [1] demonstrated that learning from hand-held gripper demonstrations also results in capable tossing skills, but has only been shown to work with *analytical controllers* on a *fixed-based* arm. For legged systems, reliable tossing depends on the dynamic body coordination. Thus, in order to achieve a successful toss, the controller must discover the right stance, utilize power from all of its degrees of freedom, and learn the coordination required to achieve precise tracking throughout the dynamic tossing motion.

| Approach Units | Pos Err $cm\downarrow$ | Orn Err $deg\downarrow$ | Survival $\%\uparrow$ | Power $kW\downarrow$ |
|---|---|---|---|---|
| Ours (Unified) | 2.18 | **3.10** | **99.8** | 3.72 |
| Ours (Tossing) | **2.12** | 3.35 | 98.4 | 3.82 |
| (-) Preview | 3.02 | 4.23 | 93.0 | 3.95 |
| (-) World-space | 15.49 | 15.55 | 0.0 | 4.74 |
| (-) UMI Traj | 2.48 | 15.67 | 97.4 | 3.69 |
| (-) Leg Ctrl | 4.01 | 5.72 | 91.0 | 3.05 |
| IK + Fixed Leg | 11.69 | 15.30 | 0.0 | **3.02** |
| DeepWBC [8] | 22.2 | 66.22 | 0.0 | 5.92 |

Table 1: **Sim Tossing**. Averaged over 500 episodes.

**Dynamic tossing requires dynamic whole-body coordination.** Through learning to track end-effector trajectories, our whole-body controller discovered a whole-body tossing strategy (Fig 5). The strategy involved dynamic whole-body coordination, involving power from all joints during the toss, momentarily balancing on one foot/two feet, and effective usage of body mass inertia to regain balance. Overall, our system achieves 75% and 70% when using motion capture and iphone odometry, respectively.

**Effects of the interface design.** From our ablations and baseline comparisons (Table 1), we observed that training on world-frame manipulation trajectories was crucial - removing either world-frame tracking or manipulation trajectories ((-) Task-space and (-) UMI Traj) significantly hurts position and/or orientation precision. Meanwhile, removing trajectory information from the WBC's observation prevents the controller from anticipating future motions, resulting in dangerous reaching behaviors ($-5.4\%$ survival) and more jittery actions ($+130W$) compared to our approach, as demonstrated in Fig 5.

**Improved tracking from diverse UMI trajectories.** When finetuned on additional trajectories from pushing and tossing, we observe that our tossing-pretrained WBC not only retained its tracking abilities, but

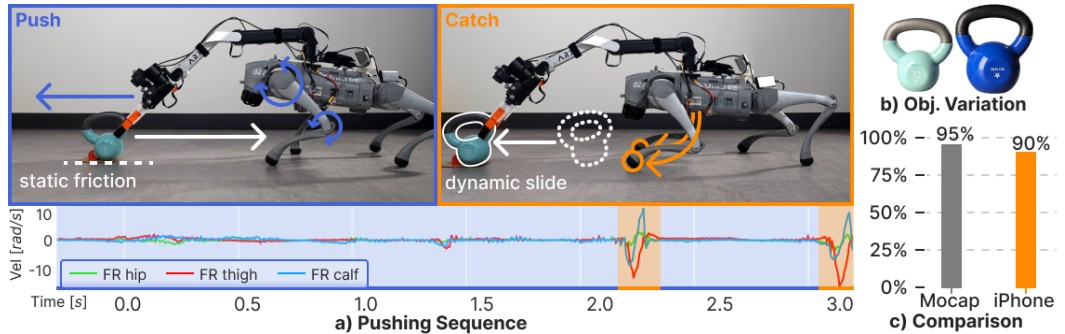

Fig 6: **Robustness to Unexpected Dynamics.** When faced with unexpected resistance from the kettlebell's friction, our controller continuously tries to push forwards. This leads to a wiggling motion along with increased pushing force until static friction breaks into a dynamic slide, at which point the robot catches itself by dropping its front feet forwards. We observed this push-and-catch strategy while the robot pushes both 10lbs and 20lbs (b), despite the latter succumbing to static friction more often. We also report comparable performance with the motion capture oracle (c).

also learned safer behavior. Quantitatively, its collision rate (1 - survival) improves over the tossing only WBC by 25% (from 1.6% to 1.2%, Table 1) while reducing its power usage by 100$W$.

**Failure analysis.** Despite the highly-dynamic nature of this motion, the controller manages safe landings during all motion capture evaluation trials. To further investigate whether tossing misses were due to imprecise WBCs or noisy end-effector trajectories, we tested our controller with calibrated tossing trajectory replay with motion capture, which effectively acts as an "oracle" manipulation policy, and report a +10% improvement (Fig. 5). We hypothesize that scaling up tossing data collection further could close this gap.

## 4.2 Robustness: End-effector Reaching Leads to Robust Whole-body Pushing

Our second task stress tests the controller's ability to handle unexpected and unseen external dynamics in zero-shot fashion. To evaluate this capability we studied **kettlebell pushing** (Fig 6), where goal of this task is to push a 10lbs kettlebell, such that it slides upright into a goal region. Between each episode, the kettlebell's distance from its goal is varied within the range [80cm,120cm], and the robot starts off with its center 60cm behind the kettlebell with its grippers grasping the kettlebell's handle.

**Task Difficulties.** Although the end effector motion involved may be simple, the whole body controller need to adapt to significantly changed dynamic models. In order to be successful, the WBC must be robust to feet slippage, overcome static friction, and adapt to when static friction suddenly breaks into a slide. In addition, small imprecision in the pushing contact point could lead to the kettlebell toppling over.

**Robustness to large disturbances.** To our surprise, even with such a large disturbance, our controller was able to maintain balance and finish the task, without ever training on simulated forces or weights. Our controller completes the task with 95% and 90% success rates for motion capture and iPhone deployment respectively. We observed that as soon as the WBC observes a larger position tracking error caused by the weight, it changes its strategy to lean forward in order to exert more force - a strategy which would cause the robot to fall forwards in absence of the kettlebell. By sending only end effector trajectories without any base velocity, the controller can also move its legs forward to reach faraway target poses. In addition, we also tested the policy with a 20lbs kettlebell, and observed 4 successful episodes out of 5.

**Failure analysis.** The most common failure case in this task was due to the kettlebell toppling sideways. In addition, we observed that our hardware system often overheats during this task's evaluation, and requires frequent cool-down breaks. Besides obvious hardware improvement, we hypothesize training with simulated forces [10] could lead to more elegant, energy efficient pushing behavior.

## 4.3 Scalability: Plug-and-play Cross-Embodiment Manipulation Policies

In our third experiment, we validate the feasibility of deploying a pre-trained manipulation policy from prior work [1] with our WBC, without fine-tuning, demonstrating zero-shot generalization to an unseen embodiment*. We evaluate this on the **In-the-wild Cup Rearrangement Task**, where the robot must place an espresso cup on a saucer with the handle facing left. The cup and saucer are randomly positioned within

---

*Zero-shot refers to the manipulation policy, which is not trained on the target embodiment

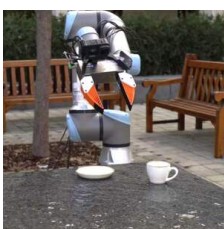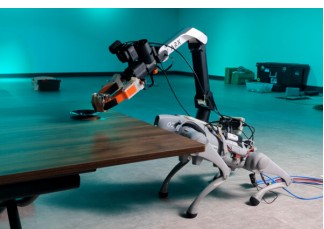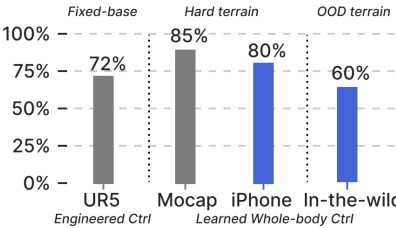

Fig 7: **Making Existing Manipulation Policies Mobile.** Although intended for fixed-based arms with larger reach range and precise controllers (left, from Chi et al.), the In-the-wild cup rearrangement policy from Chi et al. was able to achieve 80-85% success rates zero-shot on our quadruped system with our learned whole-body controller when tested on hard ground (center figure). In out-of-distribution (OOD) terrain settings (outdoor environments with deformable/unstable terrain), our system with iPhone tracking achieves 60% success rate.

a 20cm radius, with random cup orientation. The robot starts 30cm from the table edge, facing the table. Success is defined as the cup being upright on the saucer, with the handle within $\pm 15°$ of left. This task is always performed in an unseen environment for both the manipulation policy and WBC.

**Task difficulties.** The key challenge for this experiment is to effectively leverage cross-embodiment manipulation policy. In the original UMI paper [1], the policy was trained on data collected by humans, and intended for industrial robot arms (e.g., UR5 and Franka) with a fixed-base and analytical controller. In this experiment, we are directly deploying this pre-trained policy on an 18 DoF legged manipulator. Moreover, the intrinsic difficulty of the pushing and rearranging task also tests the system ability to perform precise 6 DoF movements, including prehensile and non-prehensile manipulation. This kind of precise and gentle actions have been traditional difficult for low-cost legged manipulation systems.

**Results and Findings.** Without fine-tuning the manipulation policy, our system achieved success rates of 85% and 80% using motion capture and iPhone odometry on hard flat ground, respectively, despite the cross-embodiment gap. The manipulation policy, originally designed for a UR5e system with a larger kinematic reach than our ARX5 arm, often predicts action sequences that span a larger workspace. To accommodate this, our system dynamically tilts and leans its base to increase reach and counterbalance the arm. We evaluated our system with four different cups in five locations, including various indoor and outdoor terrains like soft carpet and unstable surfaces, which were out-of-distribution (OOD) for the WBC, trained only on hard flat ground. Despite this, we achieved a 60% success rate (27/45 trials), with 72% (13/18) of failures attributed to manipulation errors (e.g., dropping or pushing the cup out of view) and 28% (5/18) due to controller shakes on OOD terrain.

### 4.4 Limitations

Although UMI-on-Legs has demonstrated various manipulation skills, several limitations remain that future work could address. First, our system currently supports only gripper-based manipulation. A promising direction is to extend our interface to facilitate whole-body manipulation [32, 51, 52]. Second, while our embodiment-agnostic interface is scalable, incorporating embodiment-specific constraints into the high-level manipulation policy is crucial. Lastly, for a complete mobile manipulation platform, our system could benefit from features like collision avoidance [53], force feedback, and force-based control [10].

## 5 Conclusion

We introduce UMI-on-Legs, a framework that combines real-world human demonstrations and simulation-based reinforcement learning. With world-frame end-effector trajectories as the interface, our formulation allows intuitive demonstrations from hand-held devices yet is expressive enough to represent complex manipulation skills such as dynamic tossing and pushing, and precise pick and place.

Although, as community, we've yet to have a unanimous foundation manipulation policy, we believe our framework is an important step towards enabling the transfer of such policies to different robot platforms. Looking forward, we're optimistic that tackling limitations in our interface (§ 4.4) could take us even closer towards a future where separate people can develop ever more general visuo-motor policies and robust WBCs, knowing that they can be plugged-and-played together in the end.

## 6  Acknowledgements

We would like to Pete Florence and Jonathan Tompson for many fruitful discussions in the project. We thank Cheng Chi, Zhenjia Xu, and Chuer Pan for guidance with UMI, and Haochen Shi, Antonio Loquercio, Haoran He, and Jiyuan Shi for tips with real-world deployment of RL controllers. A special thanks to Zhenjia Xu, for giving us the idea and starter code to try out iPhone ARKit camera pose estimation. We thank Yifan Hou, Qi Wu, Shuo Yang, Tianyu Li, Xuxin Cheng, and Kexin Shi for fruitful discussions about controller formulation, implementation and deployment. We thank Jingyun Yang and Zi-ang Cao for helpful instructions to set up the motion capture system. For paper writing and figures, we thank Zeyi Liu, Austin Patel, Mengda Xu, Dominik Bauer, Xiaomeng Xu, Mandi Zhao, and Samir Gadre for helpful feedback. Finally, we thank Carl Vondrick for providing the compute. This work was supported in part by the Google Research Award, NSF Award #2132519, #2143601, #2037101. We would like to thank ARX for the ARX robot hardware. The views and conclusions contained herein are those of the authors and should not be interpreted as necessarily representing the official policies, either expressed or implied of the sponsors.

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

# 7    Things that did not work

In this section, we discuss things we tried, but did not improve the performance or caused other issues. We hope that through openly discussing our unsuccessful attempts, the community could gain a more complete understanding of the project, learn from our mistakes, and improve upon our attempts. To complement this section, we've also included robot failure videos on our website.

## 7.1    Privileged policy distillation and observation history.

We tried to include privileged information (kp, kd, friction, damping, ground truth poses, base velocities, etc.) to train a privileged policy and distill it through supervised learning and online regularization. However, it didn't provide performance boost and introduce instability when we include observation history longer than 1 step. This could be because that the Python ROS2 timer is imprecise, thus the observation history can easily be out of distribution given the wrong history timestamps. This was less of an issue on the Unitree A1 platform, which ships with a real-time kernel.

## 7.2    Precise grasping for tossing.

To train diffusion policies for grasping and tossing, we collected 500 episodes of human demonstration. However in the grasping phase, the small shakes from the controller results in distributional drift in visuo-motor manipulation policy. We hypothesize that robust, fully-autonomous grasping and tossing could be achieved with more data.

## 7.3    System Reliability for Fully-untethered Deployment.

Although the system is capable of various tasks in the real world without any external cables, the reliability is in general not high enough. We summarize some hardware and system level challenges we encountered as follows. Given a small base-to-arm weight ratio, the joints of the dog are likely to overheat in 10 to 30 minutes according to the robot posture and then goes to an error mode. Therefore, we needed to frequently cool down the motors and fine-tune the policy to a more energy-efficient posture. On the power front, the battery supplies different voltages depending on how full it is. When fully charged, the voltage is too high for the arm, and the setup requires a voltage adaptor. When closer to empty, the policy's behavior is more dampened.

### 7.4 Velocity integration

We used the iOS ARKit to run VIO on an iPhone 15 Pro. However, the estimated camera pose sometimes drift heavily under dynamic actions including tossing. We also measure the delay of the pose estimator. Although the Ethernet communication time (between the iPhone and the onboard Jetson) is negligible, we found the the latency from the movement to the pose update is still roughly 140ms. This latency introduced a significant Sim2Real gap during deployment, which manifests as low frequency oscillations which slowly diverges.

Using low-latency foot contacts, joint position/velocities, and IMU readings, we can estimate the base velocity for each timestamp. Using base velocity estimates from 140ms into the past, we can integrate every received iPhone pose forward in time by 140ms. However, the shaking behavior due to the latency is not addressed, which can only be mitigated by higher action rate regularization fine-tuning. As the ARKit pose estimation runs at 60Hz, we expect a better implementation could achieve shorter latency.

## 8 Deployment

In this section we elaborate important details in real-world deployment and Sim2Real transfer.

### 8.1 iPhone placement

We mounted the iPhone on the back of the robot with following considerations:1) Since the iPhone is facing back, the robot arm will not be in the view during manipulation, so ARKit's visual-inertial odometry can better track the surrounding environment. 2) iphone is mounted at a fixed angle of 60° to the back plane of the Go2, thus the camera is pointing upwards even when Go2 is slightly bending down. This increases the number of visual features in the iPhone camera, thus provides more robust tracking compared to our original design, which was a 90° mount. 3) This position is kinematically unreachable for the robot arm, so the arm will unlikely damage the iPhone.

### 8.2 Robot URDF

For our highly dynamic tasks, we observed that the domain randomization and adaptive policies [8, 9] were insufficient to bridge the gap between a heavily mis-specified model (provided by the manufacturer) and the real system, a gap that is exacerbated when performing dynamic arm movements. We found that disassembling the ARX5 arm, reweighing each component, and recomputing mass, center of mass and inertia matrices in the URDF [†] were crucial for both simulation stability as well as successfully real-world deployment.

### 8.3 Latency

While all observations and actions are perfectly synchronized in simulation, the communication and code runtime of the real-world robot system introduce a significant amount of latency in different aspects.

The most important latency comes from the robot joint state observation and the joint command execution. This includes the motor encoder readout, ROS2 communication, whole-body controller inference, action sent back to motors and being executed on motors. Since we were unable to exactly measure all the latency sources, we swept the end-to-end latency from 0ms to 30ms with 5ms interval in simulation and find that 20ms works the best in our real-world system. We observed high frequency shaking if the latency is mismatched in simulator and real-world robots.

Robot pose estimators (motion capture and iphone) also introduce latency. We observed an 8ms latency from the motion capture system and 140ms latency from the iphone ARKit. We simulated a 10ms pose latency in simulation to close the sim2real gap. We also implemented inertial-legged velocity estimation to integrate the latest pose for iPhone, but the performance didn't improve significantly. See 7.4 for detailed explanation.

We also observe latency in the Python ROS2 program. Due to the global interpreter lock, all the Python ROS2 callback functions have to run alternately, so one callback function will block the others if it takes too

---

[†]Our code, data, and checkpoints will be released.

long. We optimize the run time of all callback functions and detach the callback functions that take too long to other ROS2 nodes. This allows the joint observation update to run closer to 200Hz and policy inference closer to 50Hz. A C++ implementation with more precise timer and lower latency can achieve better performance given the same checkpoint and evaluation setup.

## 8.4 Safety

We observed that directly deploying the whole-body controller checkpoints in the simulation section led to some safety issues and addressed them by fine-tuning the checkpoints with more conservative reward schemes:

- **Shaking**: To reduce the shaking and oscillation behavior of the robot, we increased the action rate regularization. The controller will get less reward if the predicted action is farther away from the previous action. We also disabled the on board lidar that introduces shaking behavior.

- **Overheat Shutdowns**: The calf joint of the robot uses a linkage configuration rather than directly applies the output torque. Therefore, the required output torque of the motor is higher than the other joints, leading to frequent overheating and emergency shutdown. We observed that increased torque regularization during training allows the controller to run up to 30 minutes continuously[‡]

- **Unsafe Configurations**: In the simulator, the controller is likely to twist the legs to achieve higher precision with less movements, but this makes it unsafe to deploy in the real world. We added reward terms to regularize the body to a more balanced pose, so that the center of mass can roughly stay in the center.

# 9 Training

## 9.1 Manipulation Policy

In this section, we report hyperparameters used for training our manipulation diffusion policy. Visual observation and proprioception horizon means how many history image and robot states (with 0.1s interval) are used as input to the policy. Output Steps means the action length of the diffusion policy output. Execution Steps and Execution Frequency indicate how many steps of the diffusion policy output is actually executed in the real-world robot. To enable more stable end-effector trajectory updates, we add linear interpolation between the executed trajectory and the newly-updated trajectory for a duration of "Trajectory Update Smoothing Time".

## 9.2 Whole-Body Dynamic Tossing

We increased the inference and execution steps for a smoother toss, since the trajectory is less likely to update during the dynamic toss. We also increased the execution frequency to get farther toss. Please refer to Table 2 for detailed parameters.

| Hyperparameters | Values |
|---|---|
| Training Set Trajectory Number | 500 |
| Diffusion Policy Visual Observation Horizon | 2 |
| Diffusion Policy Proprioception Horizon | 4 |
| Diffusion Policy Output Steps | 64 |
| Diffusion Policy Execution Steps | 40 |
| Diffusion Policy Execution Frequency | 12Hz |
| Trajectory Update Smoothing Time | 0.1s |

Table 2: **Hyper parameter set of the dynamic tossing task.**

---

[‡]Please check out evaluation videos on our website , which shows our controller running continuously for 20-30 minutes.

### 9.3 End-effector Reaching Leads to Robust Whole-body Pushing

Since the trajectory motion is much simpler, we collected fewer human demonstrations for the diffusion policy. We increased the "Trajectory Update Smoothing Time" to prevent heavy shakes due to the trajectory update. Please refer to Table 2 for detailed parameters.

| Hyperparameters | Values |
|---|---|
| Training Set Trajectory Number | 25 |
| Diffusion Policy Visual Observation Horizon | 2 |
| Diffusion Policy Proprioception Horizon | 2 |
| Diffusion Policy Output Steps | 32 |
| Diffusion Policy Execution Steps | 10 |
| Diffusion Policy Execution Frequency | 10Hz |
| Trajectory Update Smoothing Time | 0.3s |

Table 3: **Hyper parameter set of the whole-body pushing task.**

### 9.4 Plug-and-play Cross-Embodiment Manipulation Policies

As the UMI cup rearrange task requires much higher precision but is quasi-static, we decreased the execution frequency for better stability. As mentioned in the main text, we directly use the publicly released checkpoint from Chi et al.. We did not collect any extra data for this task.

| Hyperparameters | Values |
|---|---|
| Training Set Trajectory Number | 1400 |
| Diffusion Policy Visual Observation Horizon | 1 |
| Diffusion Policy Proprioception Horizon | 2 |
| Diffusion Policy Output Steps | 16 |
| Diffusion Policy Execution Steps | 8 |
| Diffusion Policy Execution Frequency | 5Hz |
| Trajectory Update Smoothing Time | 0.1s |

Table 4: **Hyper parameter set of the UMI cup rearrangement task.**

### 9.5 In-the-wild Cup Rearrangement Evaluation

We evaluated the whole-body controller on the cup rearrangement task in the wild. We tested it in some challenging scenarios including unstable terrain (grass, dirt, collapsible table), direct sunlight which made the robot easily overheated, and unseen tables and cups to show the generalizability of the diffusion policy. Please checkout the supplementary video for more details.

### 9.6 Reward Terms

During RL training of the whole body controller, we include the following reward terms:

- **Joint Limit, Joint Acceleration, Joint Torque, Root Height, Collision, Action Rate**: We use the same definition as prior works [7, 8].

- **Body-EE Alignment**: We regularize joint 0 and joint 3 of the arm (the joints that mostly affect the yaw angle of the gripper) to stay close to their initial pose. This allows the arm to be aligned with the body.

- **Even Mass Distribution**: We regularize the standard deviation of the force of the four feet to be lower. The motors are less likely to overheat if the body mass is distributed more evenly on the four legs.

- **Feet Under Hips**: We regularize the feet's planar position to be close to that of their respective hips, for all four legs. This regularization improves the stability of the standing pose.

- **Pose Reaching**: We minimize the position error $\epsilon_{\text{pos}}$ and orientation error $\epsilon_{\text{orn}}$ to the target pose using an unified reward function: $\exp\left(-\left(\frac{\epsilon_{\text{pos}}}{\sigma_{\text{pos}}} + \frac{\epsilon_{\text{orn}}}{\sigma_{\text{orn}}}\right)\right)$. We apply $\sigma$ curriculum to the reward functions: as the error get smaller, we decrease $\sigma$ for a more peaky reward function, which encourages the controller to further reduce reaching error. $\sigma_{\text{pos}}$ is set to $[2, 0.1, 0.5, 0.1, 0.05, 0.01, 0.005]$ when the position error is smaller than $[100, 1.0, 0.8, 0.5, 0.4, 0.2, 0.1]$ respectively; $\sigma_{\text{orn}}$ is set to $[8.0, 4.0, 2.0, 1.0, 0.5]$ when the orientation error is smaller than $[100.0, 1.0, 0.8, 0.6, 0.2]$ respectively.

| Name | Weight |
|---|---|
| Joint Limit | -10 |
| Joint Acceleration | -2.5e-7 |
| Joint Torque | -1e-4 |
| Root Height | -1 |
| Collision | -1 |
| Action Rate | -0.01 |
| Body-EE Alignment | -1 |
| Even Mass Distribution | -1 |
| Feet Under Hips | -1 |
| Pose Reaching | 4 |

Table 5: **Reward terms**.

# 10 Evaluation

## 10.1 Real World Tossing

To achieve a pre-grasped state during tossing evaluations, we handed the policy the tennis ball. We count episodes where the robot falls during the toss as a failure. Finally, we measure the distance from the location where the ball first lands to the center of the bin, and report a success if it lands within 40cm of the bin's center.

## 10.2 Simulation Ablations

In each episode, a random grasp and toss end-effector trajectory is sampled from our dataset (§ 9.2), the robot is uniformly randomly initialized in the range of [-5.0cm,5.0cm] for x and y position, and [-0.1rad,0.1rad] for z orientation, and its initial joint positions are sampled uniformly within 5% of that joints' full range, centered around the default joint configuration. Position and orientation errors are averaged over all timesteps, while survival is 1 only if the policy lasts for 17 seconds without a terminal collision. Here, a terminal collision is defined as any robot part contact that is not the robot's feet or gripper.

Similar to training, evaluation includes a pose latency of 10ms and the full range of domain randomization reported in Table 6.

| Hyperparameters | Values |
|---|---|
| Init XY Position | [-0.1m,0.1m] |
| Init Z Orientation | [-0.05rad,0.05rad] |
| Joint Damping | [0.01,0.5] |
| Joint Friction | [0.0,0.05] |
| Geometry Friction | [0.1,8.0] |
| Mass Randomization | [-0.25,0.25] |
| Center of Mass Randomization | [-0.1m,0.1m] |

Table 6: **Domain Randomization Hyperparameters.**

All approaches were trained for 4000 iterations, and the performance at checkpoint 4000 is reported. For Ours (Unified), we took Ours (Tossing) and finetune it for 1000 steps on 300 trajectories, 100 from each

of the three tasks studied. Power usage report is electrical power usage based on real hardware's voltages, manufacturers' reported torque constants, and the simulation's motor torques.

## 11 Misc.

**Cost of the system.** We report the cost of our entire robot system based on the market price in Table 7, which is roughly 1/4 of other quadruped manipulation systems in [10, 12]

| Item | Cost($) |
|---|---|
| Unitree Go2 Edu Plus | 12,500.00 |
| ARX5 Robot Arm | 10,000.00 |
| GoPro Hero9 | 210.99 |
| GoPro Media Mod | 79.99 |
| GoPro Max Lens Mod | 68.69 |
| iPhone 15 Pro | 999.00 |
| Elgato Capture Card | 147.34 |
| Total | 24,006.01 |

Table 7: **System Cost**.

