# OpenReview forum: "UMI-on-Legs: Making Manipulation Policies Mobile with Manipulation-Centric Whole-body Controllers"
_robot-learning.org/CoRL/2024/Conference — CoRL 2024_

### Official Review · Reviewer_uNw1 · 2024-07-20
**Solid framework for quadruped manipulator control. Possibly under-explained and oversold.**

**Originality:** 3
**Technical Quality:** 3
**Clarity Of Presentation:** 2
**Potential Impact:** 3
**Recommendation:** 3
**Confidence:** 3

**Review:**

The proposed framework is a natural decomposition of the problem --- produce target trajectories and then track those trajectories. The decomposition itself isn't that novel from a general robotics perspective, though it may be novel for this particular robot configuration.

One claimed novelty is that the produced trajectories are in the task frame, rather than the robot body frame. The claimed advantage is that this allows for the low-level controller to adjust to body shifts during execution, rather than the trajectory planner needed to adjust. This can be important when the low-level controller is operating at a much higher frequency compared to the trajectory planner.

Another claimed novelty that is mentioned a number of times is "zero-shot cross-embodiment deployment". I found the emphasis on this point to be a bit oversold. I was confused at first by this emphasis, but it appears to mean that the target trajectories can come from a number of means and not just the robot body used at deployment. Yes, I agree, but this is very different from what "zero-shot cross-embodiment" will be interpreted by many people. Many would consider "zero-shot cross-embodiment" to mean that you take your full system and it is immediately useful on a new robot (not sure what that would even mean, but it sounds impressive).

I was unable to get a full picture of how the system was trained from the paper. In particular, the paper is not self-contained in describing how data is produced for the diffusion policy learning. It is also unclear what data is used to train the low-level policy in simulation. Is training only done on the tasks considered in experiments, or is a richer set of trajectories used for training?

It wasn't clear what was controlling the end-effector opening and closing. From the description, it appeared that the trajectory data only included the position and rotation of the end-effector.

The authors say "Notably, setting up a simulation to track these manipulation end-effector trajectories does not require setting up the manipulation task and environment."  While this is notable, I don't understand how it can lead to good performance for a wide range of manipulation tasks. For example, in simulation, a tossing motion requires very different control depending on whether the gripper is empty or holding something heavy. But from the description it did not appear such a range of situations would be part of the simulations used for training. More description of this is required to provide intuition about why this works.

Given the claimed generality of the approach and the easy of creating trajectories for new tasks, I would have expected a wider variety of tasks to be explored. Currently we only see the system evaluated on 3 tasks.

Post Rebuttal:

Thanks for the clarifications. I agree that the work is making a nice contribution and those outweigh the weaknesses. I'm raising my rating to "weak accept".

**Quality Of The Limitations Section:**

3

**Questions For Rebuttal:**

The gripper state did not appear to be part of the observation space or controllable by the learned policy. How is the gripper handled in this work?

What exactly is "survival %"? Is it the same as success? For example, in tossing one might miss the goal but still not fall over.

Why is the low-level controller able to learn object manipulation when the simulations do not appear to involve objects or simulated disturbances that would be similar to such object interactions?

Am I misunderstanding what you mean by "zero-shot cross-embodiment"?

What trajectories are used to train the low-level controller?

**Robotics Focus:**

4

**Summary Of Paper:**

The paper describes a learning-based framework for manipulation with a robot arm (gripper) mounted on a quadruped base. A diffusion policy is trained via demonstrated motions to map end-effector camera input to target position trajectories for the end-effector in the task frame.  A full-body low-level controller is then trained to map the target trajectories to joint poses for the robot. Experiments are provided on three tasks both in simulation and the real-world.

**Summary Of Recommendation:**

Overall the paper is considering an interesting problem and has developed a natural and potentially effective approach. We are shown successful sim-to-real performance on 3 tasks. Given the claims of generality, the paper would have been more convincing if a wider variety of tasks was considered. The description of the system and intuition behind the results can be improved significantly.

---

### Official Review · Reviewer_H1aK · 2024-07-21
**A practical framework well designed and well communicated**

**Originality:** 4
**Technical Quality:** 4
**Clarity Of Presentation:** 5
**Potential Impact:** 3
**Recommendation:** 3
**Confidence:** 4

**Review:**

Strengths:
- The paper tackled the learning of challenging, dynamic whole-body tossing and pushing tasks enabled by UMI's free-range data collection ability.
- The paper showcased that policies trained using UMI data in end-effector trajectories can be generalized to other robot embodiments, and the robot dog's low-level policy is good enough to bridge the gap in embodiment.
- Thanks to the whole-body controlled trained using RL, the robot learned its three-foot jumping style of tossing that balances its weight.
- The figures are thoughtfully illustrated to aid understanding and videos are well made.
- The hardware set up (including iPhone odometry) is versatile for a range of whole-body manipulation tasks. The paper also included results showing that the performance is not significant from using the mocap (ground-truth) pose tracking method.

Weakness:
- It's an integration of existing tools and methods. It showcased the ease to collect data using UMI and its use in training whole-body policy. And the idea of training a whole-body controller to achieve end-effector poses is also known.
- The experiments involved very limited object and initial scene variation.

**Quality Of The Limitations Section:**

3

**Questions For Rebuttal:**

1. What exactly is task frame? The world frame? If so, why don't you call it the world frame? I was expecting to see it described in more detail in page 4, section 3.2 line 142.
2. How do you adapt the trajectory-based policy from table-top to floor-top? For example, if you have a policy trained for tossing on the table using UMI data that a human collected while standing. How do you use that policy on the robot dog with its limited height? From the figures, it looks like the human has to kneel down to collect tossing and pushing data.
3. How much data was collected to train for each task that achieved > 70% success rate? What's the distribution of initial states (e.g. height of tables in the picking task, range of robot initial base configuration and ball poses in tossing task) in demonstrations, and how different are the initial states in testing? I see in the supplementary that grasping and tossing took 500 demonstrations. How much would performance drop if only a subset of those demonstrations is used for training?
4. Is the robot's initial base positions obtained using April Tags in the environment?
5. The low-level RL policy is only used to train to each any end-effector poses, as opposed to follow certain demonstration trajectories for tasks? Would it be helpful to include task trajectories (like tossing) in training the low-level policy?

**Robotics Focus:**

4

**Summary Of Paper:**

The paper presented a framework for training mobile manipulation policies using UMI decide that uses task-space trajectory and was tested on prehensile, non-prehensile, and dynamic actions

**Summary Of Recommendation:**

I recommend accepting the paper for its thoughtful hardware setup, choice of challenging tasks, thoroughness in experiments, and clarity in communication.

---

### Official Review · Reviewer_ZZoE · 2024-07-21

**Originality:** 4
**Technical Quality:** 3
**Clarity Of Presentation:** 5
**Potential Impact:** 3
**Recommendation:** 3
**Confidence:** 5

**Review:**

Strengths
- The task-frame tracking formulation for the RL controller is novel, intuitive, and demonstrated to be helpful for manipulation.
- The ability to use cross-embodiment data is valuable since there is much more demonstration data available for fixed-base manipulators than for legged manipulators.
- The infrastructure contributions such as ARKit VIO app and precisely calibrated URDF for the system will be valuable tools for diverse legged manipulation applications if open sourced.
- The writing is very clear and evaluations are thorough.

Weaknesses
- I couldn't find some details about the trajectories used during RL policy training.
- It’s not quantitatively evaluated to what extent, and in what aspects, the demonstrated tasks take advantage of the base having legs.
- As acknowledged in the limitations, it's not clear how to propagate embodiment-specific constraints back to the high-level manipulation policy, which would be necessary to ensure the scalability of the decoupled plug-and-play approach.

**Quality Of The Limitations Section:**

3

**Questions For Rebuttal:**

“To provide the controller with relevant reference trajectories, we use trajectories collected with UMI [1].” .. “removing either task-frame tracking or manipulation trajectories ((-) Task-space and (-) UMI Traj) significantly hurts position and/or orientation precision.” I would like more details about the trajectories used to train the RL controller. How many trajectories are there and what kinds of tasks do they represent? E.g. are the trajectories in 4.2 and 4.3 also seen during training or are they out-of-distribution for the RL policy? If I wanted to add a new task with UMI, would it also be important to retrain the WBC on some task-related trajectories, or does the policy learn a very general tracking capability?

It would be really cool to see the system accomplish tasks that specifically require legs. For example, can the robot walk over some uneven terrain (inaccessible to a wheeled robot) before performing the manipulation, while carrying the UMI payload? This could even be done by switching to Unitree’s default controller, if it’s hard to include non-flat terrain in the RL training.

Further thoughts on this topic: “For legged systems, reliable tossing depends on the body placement and the tossing motion. Thus, in order to achieve a successful toss, the controller must discover the right stance, utilize power from all of its degrees of freedom, and learn the dynamic whole-body coordination required to achieve precise tracking throughout the dynamic tossing motion.” Can this be quantified? For example, you could consider training an ablation policy (a) where the robot’s legs have fixed targets and the action space only controls the arm joints. Or you could consider an ablation (b) where the legs have fixed targets and the gripper is controlled with IK. I think it’s reasonable to expect (b) will not be performant because the base will shake — but this would be nice to show. I think (a) could actually perform better because the policy can still learn to compensate the arm’s commands for the motion of the base. If the full system delivers the best performance, the gap would quantify how much the legs are really contributing power or stabilization towards the task, vs. primarily acting as a complication to the dynamics that needs to be overcome. I think both are suggested in different parts of the paper and we don't really know which is being evidenced here.

"With an expressive embodiment-agnostic interface in place, we can separately develop ever more general visuo-motor policies and robust WBCs, knowing that they can be plugged-and-played together in the end." As mentioned in the limitations section, there is the outstanding problem of making the high-level policy aware of the embodiment-specific constraints. With this problem unsolved, can we really claim yet that developing more general versions of these components will lead to a plug-and-play solution?

**Robotics Focus:**

4

**Summary Of Paper:**

This paper proposes UMI-on-legs which integrates the Universal Manipulation Interface (UMI) from prior work with a quadruped robot base. This is enabled by substituting the task-space end-effector controllers used in UMI for a RL whole-body controller that controls the legs and arm jointly to track task-space manipulator commands.

**Summary Of Recommendation:**

The paper makes a clear contribution to the field of legged mobile manipulation. It would be enhanced by clarifying points of the training procedure and ablating claims about the role of the legged base in the realized behaviors.

---

### Author Rebuttal · Authors · 2024-08-07

Please find attached a zip file containing:
- 1. The revised PDF, with changes highlighted in pink.
- 2. The 1x speed video of all 45 evaluation rollouts of the in-the-wild cup rearrangement experiment (also available at [this link](https://corlpaper.github.io/videos/cup_in_the_wild_rollouts_small_360p.mp4)).
- 3. Videos of demo rollouts of our system walking over rough terrain, then tossing into a target bin (also available at [this link](https://corlpaper.github.io/videos/walk_and_toss/000.mp4) with more examples on the [supplementary website](https://corlpaper.github.io)).

---

### Decision · Program_Chairs · 2024-09-04

**Decision:**

Accept

**Comment:**

**Pre-rebuttal**

This paper receives a mixed review from three reviewers.

The reviews acknowledge the following strengths:
- Effectiveness of the whole-body controller trained with RL (**ZZoE**, **H1aK**).
- Showcasing a way of leveraging hand-held gripper data for quadruped manipulator (**ZZoE**, **H1aK**).
- Versatility of hardware setup (**H1aK**).
- Clarity of presentation (**ZZoE**, **H1aK**).
- Potential infrastructure contribution if open sourced (**ZZoE**).

The following weaknesses are mentioned:
- Lack of details on training (**ZZoE**, **uNw1**).
- Insufficient demonstration on tasks that require a legged base (**ZZoE**).
- Concern and confusion on the "zero-shot cross-embodiment" claim (**ZZoE**, **uNw1**).
- Limited technical novelty (**H1aK**).
- Lacking on the evaluated task number and variation (**H1aK**, **uNw1**).

---
**Post-rebuttal**

The paper initially received two weak accept and one weak reject. The reviewer gave the weak reject acknowledged the merit of the rebuttal and raised the assessment to weak accept. This leads to a positive consensus among three reviewers.

Overall the paper has delivered a novel and interesting framework for combining quadruped whole-body control with task frame human demonstrations. AC agrees with the assessments of the reviewers and thereby recommends accept.